# Adaptive Cross-Modal Few-shot Learning

**Chen Xing**[*]
College of Computer Science,
Nankai University, Tianjin, China
Element AI, Montreal, Canada

**Negar Rostamzadeh**
Element AI, Montreal, Canada

**Boris N. Oreshkin**
Element AI, Montreal, Canada

**Pedro O. Pinheiro**
Element AI, Montreal, Canada

## Abstract

Metric-based meta-learning techniques have successfully been applied to few-shot classification problems. In this paper, we propose to leverage cross-modal information to enhance metric-based few-shot learning methods. Visual and semantic feature spaces have different structures by definition. For certain concepts, visual features might be richer and more discriminative than text ones. While for others, the inverse might be true. Moreover, when the support from visual information is limited in image classification, semantic representations (learned from unsupervised text corpora) can provide strong prior knowledge and context to help learning. Based on these two intuitions, we propose a mechanism that can adaptively combine information from both modalities according to new image categories to be learned. Through a series of experiments, we show that by this adaptive combination of the two modalities, our model outperforms current uni-modality few-shot learning methods and modality-alignment methods by a large margin on all benchmarks and few-shot scenarios tested. Experiments also show that our model can effectively adjust its focus on the two modalities. The improvement in performance is particularly large when the number of shots is very small.

## 1 Introduction

Deep learning methods have achieved major advances in areas such as speech, language and vision [25]. These systems, however, usually require a large amount of labeled data, which can be impractical or expensive to acquire. Limited labeled data lead to overfitting and generalization issues in classical deep learning approaches. On the other hand, existing evidence suggests that human visual system is capable of effectively operating in small data regime: humans can learn new concepts from a very few samples, by leveraging prior knowledge and context [23, 30, 46]. The problem of learning new concepts with small number of labeled data points is usually referred to as *few-shot learning* [1, 6, 27, 22] (FSL).

Most approaches addressing few-shot learning are based on *meta-learning* paradigm [43, 3, 52, 13], a class of algorithms and models focusing on learning how to (quickly) learn new concepts. Meta-learning approaches work by learning a parameterized function that embeds a variety of learning tasks and can generalize to new ones. Recent progress in few-shot image classification has primarily been made in the context of unimodal learning. In contrast to this, employing data from another modality can help when the data in the original modality is limited. For example, strong evidence supports the hypothesis that language helps recognizing new visual objects in toddlers [15, 45]. This

---

[*]Work done when interning at Element AI. Contact through: xingchen1113@gmail.com

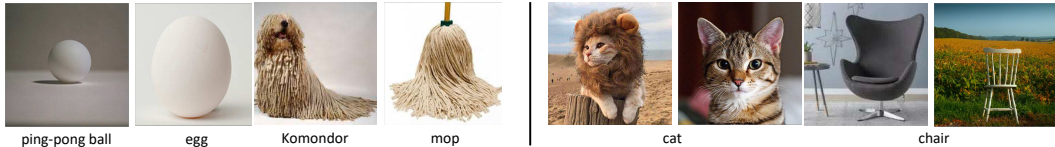

Figure 1: Concepts have different visual and semantic feature space. *(Left)* Some categories may have similar visual features and dissimilar semantic features. *(Right)* Other can possess same semantic label but very distinct visual features. Our method adaptively exploits both modalities to improve classification performance in low-shot regime.

suggests that semantic features from text can be a powerful source of information in the context of few-shot image classification.

Exploiting auxiliary modality (*e.g.*, attributes, unlabeled text corpora) to help image classification when data from visual modality is limited, have been mostly driven by *zero-shot learning* [24, 36] (ZSL). ZSL aims at recognizing categories whose instances have not been seen during training. In contrast to few-shot learning, there is no small number of labeled samples from the original modality to help recognize new categories. Therefore, most approaches consist of aligning the two modalities during training. Through this *modality-alignment*, the modalities are mapped together and forced to have the same semantic structure. This way, knowledge from auxiliary modality is transferred to the visual side for new categories at test time [9].

However, visual and semantic feature spaces have heterogeneous structures by definition. For certain concepts, visual features might be richer and more discriminative than text ones. While for others, the inverse might be true. Figure 1 illustrates this remark. Moreover, when the number of support images from visual side is very small, information provided from this modality tend to be noisy and local. On the contrary, semantic representations (learned from large unsupervised text corpora) can act as more general prior knowledge and context to help learning. Therefore, instead of aligning the two modalities (to transfer knowledge to the visual modality), for few-shot learning in which information are provided from both modalities during test, it is better to treat them as two independent knowledge sources and adaptively exploit both modalities according to different scenarios. Towards this end, we propose *Adaptive Modality Mixture Mechanism* (AM3), an approach that adaptively and selectively combines information from two modalities, visual and semantic, for few-shot learning.

AM3 is built on top of metric-based meta-learning approaches. These approaches perform classification by comparing distances in a learned metric space (from visual data). On the top of that, our method also leverages text information to improve classification accuracy. AM3 performs classification in an adaptive convex combination of the two distinctive representation spaces with respect to image categories. With this mechanism, AM3 can leverage the benefits from both spaces and adjust its focus accordingly. For cases like Figure 1(Left), AM3 focuses more on the semantic modality to obtain general context information. While for cases like Figure 1(Right), AM3 focuses more on the visual modality to capture rich local visual details to learn new concepts.

Our main contributions can be summarized as follows: (i) we propose adaptive modality mixture mechanism (AM3) for cross-modal few-shot classification. AM3 adapts to few-shot learning better than modality-alignment methods by adaptively mixing the semantic structures of the two modalities. (ii) We show that our method achieves considerable boost in performance over different metric-based meta-learning approaches. (iii) AM3 outperforms by a considerable margin current (single-modality and cross-modality) state of the art in few-shot classification on different datasets and different number of shots. (iv) We perform quantitative investigations to verify that our model can effectively adjust its focus on the two modalities according to different scenarios.

## 2  Related Work

**Few-shot learning.**  Meta-learning has a prominent history in machine learning [43, 3, 52]. Due to advances in representation learning methods [11] and the creation of new few-shot learning datasets [22, 53], many deep meta-learning approaches have been applied to address the few-shot learning problem . These methods can be roughly divided into two main types: metric-based and gradient-based approaches.

Metric-based approaches aim at learning representations that minimize intra-class distances while maximizing the distance between different classes. These approaches rely on an episodic training

framework: the model is trained with sub-tasks (episodes) in which there are only a few training samples for each category. For example, matching networks [53] follows a simple nearest neighbour framework. In each episode, it uses an attention mechanism (over the encoded support) as a similarity measure for one-shot classification.

In prototypical networks [47], a metric space is learned where embeddings of queries of one category are close to the centroid (or prototype) of supports of the same category, and far away from centroids of other classes in the episode. Due to the simplicity and good performance of this approach, many methods extended this work. For instance, Ren *et al.* [39] propose a semi-supervised few-shot learning approach and show that leveraging unlabeled samples outperform purely supervised prototypical networks. Wang et al. [54] propose to augment the support set by generating hallucinated examples. Task-dependent adaptive metric (TADAM) [35] relies on conditional batch normalization [5] to provide task adaptation (based on task representations encoded by visual features) to learn a task-dependent metric space.

Gradient-based meta-learning methods aim at training models that can generalize well to new tasks with only a few fine-tuning updates. Most these methods are built on top of model-agnostic meta-learning (MAML) framework [7]. Given the universality of MAML, many follow-up works were recently proposed to improve its performance on few-shot learning [33, 21]. Kim *et al.* [18] and *Finn et al.* [8] propose a probabilistic extension to MAML trained with variational approximation. Conditional class-aware meta-learning (CAML) [16] conditionally transforms embeddings based on a metric space that is trained with prototypical networks to capture inter-class dependencies. Latent embedding optimization (LEO) [41] aims to tackle MAML's problem of only using a few updates on a low data regime to train models in a high dimensional parameter space. The model employs a low-dimensional latent model embedding space for update and then decodes the actual model parameters from the low-dimensional latent representations. This simple yet powerful approach achieves current state of the art result in different few-shot classification benchmarks. Other meta-learning approaches for few-shot learning include using memory architecture to either store exemplar training samples [42] or to directly encode fast adaptation algorithm [38]. Mishra et al. [32] use temporal convolution to achieve the same goal.

Current approaches mentioned above rely solely on visual features for few-shot classification. Our contribution is orthogonal to current metric-based approaches and can be integrated into them to boost performance in few-shot image classification.

**Zero-shot learning.** Current ZSL methods rely mostly on visual-auxiliary modality alignment [9, 58]. In these methods, samples for the same class from the two modalities are mapped together so that the two modalities obtain the same semantic structure. There are three main families of modality alignment methods: representation space alignment, representation distribution alignment and data synthetic alignment.

Representation space alignment methods either map the visual representation space to the semantic representation space [34, 48, 9], or map the semantic space to the visual space [59]. Distribution alignment methods focus on making the alignment of the two modalities more robust and balanced to unseen data [44]. ReViSE [14] minimizes maximum mean discrepancy (MMD) of the distributions of the two representation spaces to align them. CADA-VAE [44] uses two VAEs [19] to embed information for both modalities and align the distribution of the two latent spaces. Data synthetic methods rely on generative models to generate image or image feature as data augmentation [60, 57, 31, 54] for unseen data to train the mapping function for more robust alignment.

ZSL does not have access to any visual information when learning new concepts. Therefore, ZSL models have no choice but to align the two modalities. This way, during test the image query can be directly compared to auxiliary information for classification [59]. Few-shot learning, on the other hand, has access to a small amount of support images in the original modality during test. This makes alignment methods from ZSL seem unnecessary and too rigid for FSL. For few-shot learning, it would be better if we could preserve the distinct structures of both modalities and adaptively combine them for classification according to different scenarios. In Section 4 we show that by doing so, AM3 outperforms directly applying modality alignment methods for few-shot learning by a large margin.

# 3 Method

In this section, we explain how AM3 adaptively leverages text data to improve few-shot image classification. We start with a brief explanation of episodic training for few-shot learning and a

summary of prototypical networks followed by the description of the proposed adaptive modality mixture mechanism.

## 3.1 Preliminaries

### 3.1.1 Episodic Training

Few-shot learning models are trained on a labeled dataset $\mathcal{D}_{\text{train}}$ and tested on $\mathcal{D}_{\text{test}}$. The class sets are disjoint between $\mathcal{D}_{\text{train}}$ and $\mathcal{D}_{\text{test}}$. The test set has only a few labeled samples per category. Most successful approaches rely on an *episodic* training paradigm: the few shot regime faced at test time is simulated by sampling small samples from the large labeled set $\mathcal{D}_{\text{train}}$ during training.

In general, models are trained on $K$-shot, $N$-way episodes. Each episode $e$ is created by first sampling $N$ categories from the training set and then sampling two sets of images from these categories: (i) the *support* set $\mathcal{S}_e = \{(s_i, y_i)\}_{i=1}^{N \times K}$ containing $K$ examples for each of the $N$ categories and (ii) the *query* set $\mathcal{Q}_e = \{(q_j, y_j)\}_{j=1}^{Q}$ containing different examples from the same $N$ categories.

The episodic training for few-shot classification is achieved by minimizing, for each episode, the loss of the prediction on samples in query set, given the support set. The model is a parameterized function and the loss is the negative loglikelihood of the true class of each query sample:

$$\mathcal{L}(\theta) = \underset{(\mathcal{S}_e, \mathcal{Q}_e)}{\mathbb{E}} - \sum_{t=1}^{Q_e} \log p_\theta(y_t | q_t, \mathcal{S}_e) \,, \tag{1}$$

where $(q_t, y_t) \in \mathcal{Q}_e$ and $\mathcal{S}_e$ are, respectively, the sampled query and support set at episode $e$ and $\theta$ are the parameters of the model.

### 3.1.2 Prototypical Networks

We build our model on top of metric-based meta-learning methods. We choose prototypical network [47] for explaining our model due to its simplicity. We note, however, that the proposed method can potentially be applied to any metric-based approach.

Prototypical networks use the support set to compute a centroid (prototype) for each category (in the sampled episode) and query samples are classified based on the distance to each prototype. The model is a convolutional neural network [26] $f : \mathbb{R}^{n_v} \to \mathbb{R}^{n_p}$, parameterized by $\theta_f$, that learns a $n_p$-dimensional space where samples of the same category are close and those of different categories are far apart.

For every episode $e$, each embedding prototype $p_c$ (of category $c$) is computed by averaging the embeddings of all support samples of class $c$:

$$\mathbf{p}_c = \frac{1}{|S_e^c|} \sum_{(s_i, y_i) \in \mathcal{S}_e^c} f(s_i) \,, \tag{2}$$

where $\mathcal{S}_e^c \subset \mathcal{S}_e$ is the subset of support belonging to class $c$.

The model produces a distribution over the $N$ categories of the episode based on a softmax [4] over (negative) distances $d$ of the embedding of the query $q_t$ (from category $c$) to the embedded prototypes:

$$p(y = c | q_t, S_e, \theta) = \frac{\exp(-d(f(q_t), \mathbf{p}_c))}{\sum_k \exp(-d(f(q_t), \mathbf{p}_k))} \,. \tag{3}$$

We consider $d$ to be the Euclidean distance. The model is trained by minimizing Equation 1 and the parameters are updated with stochastic gradient descent.

## 3.2 Adaptive Modality Mixture Mechanism

The information contained in semantic concepts can significantly differ from visual contents. For instance, 'Siberian husky' and 'wolf', or 'komondor' and 'mop', might be difficult to discriminate with visual features, but might be easier to discriminate with language semantic features.

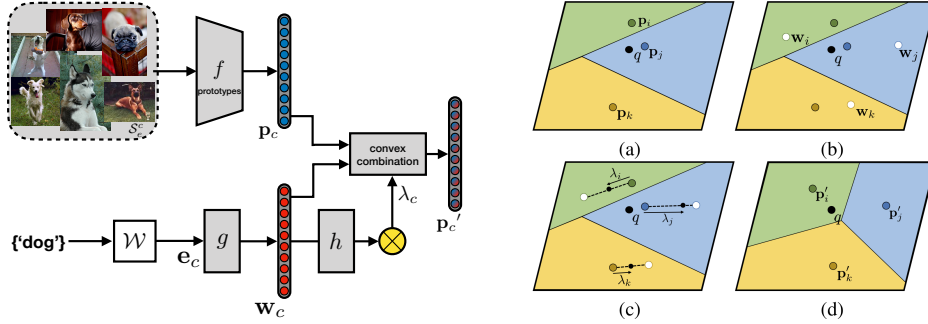

Figure 2: *(Left)* Adaptive modality mixture model. The final category prototype is a convex combination of the visual and the semantic feature representations. The mixing coefficient is conditioned on the semantic label embedding. *(Right)* Qualitative example of how AM3 works. Assume query sample $q$ has category $i$. (a) The closest visual prototype to the query sample $q$ is $\mathbf{p}_j$. (b) The semantic prototypes. (c) The mixture mechanism modify the positions of the prototypes, given the semantic embeddings. (d) After the update, the closest prototype to the query is now the one of the category $i$, correcting the classification.

In zero-shot learning, where no visual information is given at test time (that is, the support set is void), algorithms need to solely rely on an auxiliary (*e.g.*, text) modality. On the other extreme, when the number of labeled image samples is large, neural network models tend to ignore the auxiliary modality as it is able to generalize well with large number of samples [20].

Few-shot learning scenario fits in between these two extremes. Thus, we hypothesize that both visual and semantic information can be useful for few-shot learning. Moreover, given that visual and semantic spaces have different structures, it is desirable that the proposed model exploits both modalities adaptively, given different scenarios. For example, when it meets objects like 'ping-pong balls' which has many visually similar counterparts, or when the number of shots is very small from the visual side, it relies more on text modality to distinguish them.

In AM3, we augment metric-based FSL methods to incorporate language structure learned by a word-embedding model $\mathcal{W}$ (pre-trained on unsupervised large text corpora), containing label embeddings of all categories in $\mathcal{D}_{\text{train}} \cup \mathcal{D}_{\text{test}}$. In our model, we modify the prototype representation of each category by taking into account their label embeddings.

More specifically, we model the new prototype representation as a convex combination of the two modalities. That is, for each category $c$, the new prototype is computed as:

$$\mathbf{p}'_c = \lambda_c \cdot \mathbf{p}_c + (1 - \lambda_c) \cdot \mathbf{w}_c \,, \tag{4}$$

where $\lambda_c$ is the *adaptive mixture coefficient* (conditioned on the category) and $\mathbf{w}_c = g(\mathbf{e}_c)$ is a transformed version of the label embedding for class $c$. The representation $\mathbf{e}_c$ is the pre-trained word embedding of label $c$ from $\mathcal{W}$. This transformation $g : \mathbb{R}^{n_w} \to \mathbb{R}^{n_p}$, parameterized by $\theta_g$, is important to guarantee that both modalities lie on the space $\mathbb{R}^{n_p}$ of the same dimension and can be combined. The coefficient $\lambda_c$ is conditioned on category and calculated as follows:

$$\lambda_c = \frac{1}{1 + \exp(-h(\mathbf{w}_c))} \,, \tag{5}$$

where $h$ is the adaptive mixing network, with parameters $\theta_h$. Figure 2(left) illustrates the proposed model. The mixing coefficient $\lambda_c$ can be conditioned on different variables. In Appendix F we show how performance changes when the mixing coefficient is conditioned on different variables.

The training procedure is similar to that of the original prototypical networks. However, the distances $d$ (used to calculate the distribution over classes for every image query) are between the query and the cross-modal prototype $\mathbf{p}'_c$:

$$p_\theta(y = c | q_t, S_e, \mathcal{W}) = \frac{\exp(-d(f(q_t), \mathbf{p}'_c))}{\sum_k \exp(-d(f(q_t), \mathbf{p}'_k))} \,, \tag{6}$$

where $\theta = \{\theta_f, \theta_g, \theta_h\}$ is the set of parameters. Once again, the model is trained by minimizing Equation 1. Note that in this case the probability is also conditioned on the word embeddings $\mathcal{W}$.

Figure 2(right) illustrates an example on how the proposed method works. Algorithm 1, on supplementary material, shows the pseudocode for calculating the episode loss. We chose prototypical network [47] for explaining our model due to its simplicity. We note, however, that AM3 can potentially be applied to any metric-based approach that calculates prototypical embeddings $\mathbf{p}_c$ for categories. As shown in next section, we apply AM3 on both ProtoNets and TADAM [35]. TADAM is a task-dependent metric-based few-shot learning method, which currently performs the best among all metric-based FSL methods.

## 4 Experiments

In this section we compare our model, AM3, with three different types of baselines: uni-modality few-shot learning methods, modality-alignment methods and metric-based extensions of modality-alignment methods. We show that AM3 outperforms the state of the art of each family of baselines. We also verify the adaptiveness of AM3 through quantitative analysis.

### 4.1 Experimental Setup

We conduct main experiments with two widely used few-shot learning datasets: *mini*ImageNet [53] and *tiered*ImageNet [39]. We also experiment on CUB-200 [55], a widely used zero-shot learning dataset. We evaluate on this dataset to provide a more direct comparison with modality-alignment methods. This is because most modality-alignment methods have no published results on few-shot datasets. We use GloVe [37] to extract the word embeddings for the category labels of the two image few-shot learning data sets. The embeddings are trained with large unsupervised text corpora.

More details about the three datasets can be found in Appendix B.

**Baselines.** We compare AM3 with three family of methods. The first is uni-modality few-shot learning methods such as MAML [7], LEO [41], Prototypical Nets [47] and TADAM [35]. LEO achieves current state of the art among uni-modality methods. The second fold is modality alignment methods. CADA-VAE [44], among them, has the best published results on both zero and few-shot learning. To better extend modality alignment methods to few-shot setting, we also apply the metric-based loss and the episode training of ProtoNets on their visual side to build a visual representation space that better fits few-shot scenario. This leads to the third fold baseline, modality alignment methods extended to metric-based FSL.

Details of baseline implementations can be found in Appendix C.

**AM3 Implementation.** We test AM3 with two backbone metric-based few-shot learning methods: ProtoNets and TADAM. In our experiments, we use the stronger ProtoNets implementation of [35], which we call ProtoNets++. Prior to AM3, TADAM achieves the current state of the art among all metric-based few-shot learning methods. For details on network architectures, training and evaluation procedures, see Apprendix D. Source code is released at *https://github.com/ElementAI/am3*.

### 4.2 Results

Table 1 and Table 2 show classification accuracy on *mini*ImageNet and on *tiered*ImageNet, respectively. We conclude multiple results from these experiments. First, AM3 outperforms its backbone methods by a large margin in all cases tested. This indicates that when properly employed, text modality can be used to boost performance in metric-based few-shot learning framework very effectively.

Second, AM3 (with TADAM backbone) achieves results superior to current state of the art (in both single modality FSL and modality alignment methods). The margin in performance is particularly remarkable in the 1-shot scenario. The margin of AM3 w.r.t. uni-modality methods is larger with smaller number of shots. This indicates that the lower the visual content is, the more important semantic information is for classification. Moreover, the margin of AM3 w.r.t. modality alignment methods is larger with smaller number of shots. This indicates that the adaptiveness of AM3 would be more effective when the visual modality provides less information. A more detailed analysis about the adaptiveness of AM3 is provided in Section 4.3.

| Model | Test Accuracy | | |
|---|---|---|---|
| | 5-way 1-shot | 5-way 5-shot | 5-way 10-shot |
| *Uni-modality few-shot learning baselines* | | | |
| Matching Network [53] | $43.56 \pm 0.84\%$ | $55.31 \pm 0.73\%$ | - |
| Prototypical Network [47] | $49.42 \pm 0.78\%$ | $68.20 \pm 0.66\%$ | $74.30 \pm 0.52\%$ |
| Discriminative k-shot [2] | $56.30 \pm 0.40\%$ | $73.90 \pm 0.30\%$ | $78.50 \pm 0.00\%$ |
| Meta-Learner LSTM [38] | $43.44 \pm 0.77\%$ | $60.60 \pm 0.71\%$ | - |
| MAML [7] | $48.70 \pm 1.84\%$ | $63.11 \pm 0.92\%$ | - |
| ProtoNets w Soft k-Means [39] | $50.41 \pm 0.31\%$ | $69.88 \pm 0.20\%$ | - |
| SNAIL [32] | $55.71 \pm 0.99\%$ | $68.80 \pm 0.92\%$ | - |
| CAML [16] | $59.23 \pm 0.99\%$ | $72.35 \pm 0.71\%$ | - |
| LEO [41] | $61.76 \pm 0.08\%$ | $77.59 \pm 0.12\%$ | - |
| *Modality alignment baselines* | | | |
| DeViSE [9] | $37.43\pm0.42\%$ | $59.82\pm0.39\%$ | $66.50\pm0.28\%$ |
| ReViSE [14] | $43.20\pm0.87\%$ | $66.53\pm0.68\%$ | $72.60\pm0.66\%$ |
| CBPL [29] | $58.50\pm0.82\%$ | $75.62\pm0.61\%$ | - |
| f-CLSWGAN [57] | $53.29\pm0.82\%$ | $72.58\pm0.27\%$ | $73.49\pm0.29\%$ |
| CADA-VAE [44] | $58.92\pm1.36\%$ | $73.46\pm1.08\%$ | $76.83\pm0.98\%$ |
| *Modality alignment baselines extended to metric-based FSL framework* | | | |
| DeViSE-FSL | $56.99 \pm 1.33\%$ | $72.63 \pm 0.72\%$ | $76.70 \pm 0.53\%$ |
| ReViSE-FSL | $57.23 \pm 0.76\%$ | $73.85 \pm 0.63\%$ | $77.21 \pm 0.31\%$ |
| f-CLSWGAN-FSL | $58.47 \pm 0.71\%$ | $72.23 \pm 0.45\%$ | $76.90 \pm 0.38\%$ |
| CADA-VAE-FSL | $61.59 \pm 0.84\%$ | $75.63 \pm 0.52\%$ | $79.57 \pm 0.28\%$ |
| *AM3 and its backbones* | | | |
| ProtoNets++ | $56.52 \pm 0.45\%$ | $74.28 \pm 0.20\%$ | $78.31 \pm 0.44\%$ |
| AM3-ProtoNets++ | $65.21 \pm 0.30\%$ | $75.20 \pm 0.27\%$ | $78.52 \pm 0.28\%$ |
| TADAM [35] | $58.56 \pm 0.39\%$ | $76.65 \pm 0.38\%$ | $80.83 \pm 0.37\%$ |
| AM3-TADAM | $\mathbf{65.30 \pm 0.49\%}$ | $\mathbf{78.10 \pm 0.36\%}$ | $\mathbf{81.57 \pm 0.47\%}$ |

Table 1: Few-shot classification accuracy on *test* split of *mini*ImageNet. Results in the top use only visual features. Modality alignment baselines are shown on the middle and our results (and their backbones) on the bottom part.

| Model | Test Accuracy | |
|---|---|---|
| | 5-way 1-shot | 5-way 5-shot |
| *Uni-modality few-shot learning baselines* | | |
| MAML[†] [7] | $51.67 \pm 1.81\%$ | $70.30 \pm 0.08\%$ |
| Proto. Nets with Soft k-Means [39] | $53.31 \pm 0.89\%$ | $72.69 \pm 0.74\%$ |
| Relation Net[†] [50] | $54.48 \pm 0.93\%$ | $71.32 \pm 0.78\%$ |
| Transductive Prop. Nets [28] | $54.48 \pm 0.93\%$ | $71.32 \pm 0.78\%$ |
| LEO [41] | $66.33 \pm 0.05\%$ | $81.44 \pm 0.09\%$ |
| *Modality alignment baselines* | | |
| DeViSE [9] | $49.05\pm0.92\%$ | $68.27\pm0.73\%$ |
| ReViSE [14] | $52.40\pm0.46\%$ | $69.92\pm0.59\%$ |
| CADA-VAE [44] | $58.92\pm1.36\%$ | $73.46\pm1.08\%$ |
| *Modality alignment baselines extended to metric-based FSL framework* | | |
| DeViSE-FSL | $61.78 \pm 0.43\%$ | $77.17 \pm 0.81\%$ |
| ReViSE-FSL | $62.77 \pm 0.31\%$ | $77.27 \pm 0.42\%$ |
| CADA-VAE-FSL | $63.16 \pm 0.93\%$ | $78.86 \pm 0.31\%$ |
| *AM3 and its backbones* | | |
| ProtoNets++ | $58.47 \pm 0.64\%$ | $78.41 \pm 0.41\%$ |
| AM3-ProtoNets++ | $67.23 \pm 0.34\%$ | $78.95 \pm 0.22\%$ |
| TADAM [35] | $62.13 \pm 0.31\%$ | $81.92 \pm 0.30\%$ |
| AM3-TADAM | $\mathbf{69.08 \pm 0.47\%}$ | $\mathbf{82.58 \pm 0.31\%}$ |

Table 2: Few-shot classification accuracy on *test* split of *tiered*ImageNet. Results in the top use only visual features. Modality alignment baselines are shown in the middle and our results (and their backbones) in the bottom part. [†]deeper net, evaluated in [28].

Finally, it is also worth noting that all modality alignment baselines get a significant performance improvement when extended to metric-based, episodic, few-shot learning framework. However, most

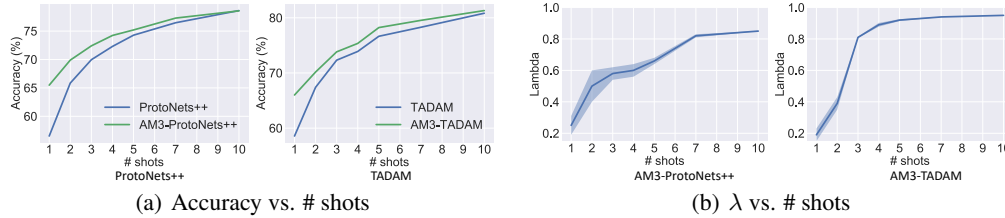

(a) Accuracy vs. # shots           (b) $\lambda$ vs. # shots

Figure 3: (a) Comparison of AM3 and its corresponding backbone for different number of shots (b) Average value of $\lambda$ (over whole validation set) for different number of shot, considering both backbones.

of modality alignment methods (original and extended), perform worse than current state-of-the-art uni-modality few-shot learning method. This indicates that although modality alignment methods are effective for cross-modality in ZSL, it does not fit few-shot scenario very much. One possible reason is that when aligning the two modalities, some information from both sides could be lost because two distinct structures are forced to align.

We also conducted few-shot learning experiments on CUB-200, a popular dataest for ZSL dataset, to better compare with published results of modality alignment methods. All the conclusion discussed above hold true on CUB-200. Moreover, we also conduct ZSL and generalized FSL experiments to verify the importance of the proposed adaptive mechanism. Results on on this dataset are shown in Appendix E.

## 4.3 Adaptiveness Analysis

We argue that the adaptive mechanism is the main reason for the performance boosts observed in the previous section. We design an experiment to quantitatively verify that the adaptive mechanism of AM3 can adjust its focus on the two modalities reasonably and effectively.

Figure 3(a) shows the accuracy of our model compared to the two backbones tested (ProtoNets++ and TADAM) on *mini*ImageNet for 1-10 shot scenarios. It is clear from the plots that the gap between AM3 and the corresponding backbone gets reduced as the number of shots increases. Figure 3(b) shows the mean and std (over whole validation set) of the mixing coefficient $\lambda$ for different shots and backbones.

First, we observe that the mean of $\lambda$ correlates with number of shots. This means that AM3 weighs more on text modality (and less on visual one) as the number of shots (hence, the number of visual data points) decreases. This trend suggests that AM3 can automatically adjust its focus more to text modality to help classification when information from the visual side is very low. Second, we can also observe that the variance of $\lambda$ (shown in Figure 3(b)) correlates with the performance gap of AM3 and its backbone methods (shown in Figure 3(a)). When the variance of $\lambda$ decreases with the increase of number of shots, the performance gap also shrinks. This indicates that the adaptiveness of AM3 on category level plays a very important role for the performance boost.

## 5 Conclusion

In this paper, we propose a method that can adaptively and effectively leverage cross-modal information for few-shot classification. The proposed method, AM3, boosts the performance of metric-based approaches by a large margin on different datasets and settings. Moreover, by leveraging unsupervised textual data, AM3 outperforms state of the art on few-shot classification by a large margin. The textual semantic features are particularly helpful on the very low (visual) data regime (*e.g.* one-shot). We also conduct quantitative experiments to show that AM3 can reasonably and effectively adjust its focus on the two modalities.

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
