[Supplementary Material]

## A  Algorithm for Episode Loss

Algorithm 1: Training episode loss computation for adaptive cross-modality few-shot learning. $M$ is the total number of classes in the training set, $N$ is the number of classes in every episode, $K$ is the number of supports for each class, $K_Q$ is the number of queries for each class, $\mathcal{W}$ is the pretrained label embedding dictionary.

**Input**: Training set $\mathcal{D}_{\text{train}} = \{(\mathbf{x}_i, y_i)\}_i, y_i \in \{1, ..., M\}$. $\mathcal{D}_{\text{train}}^c = \{(\mathbf{x}_i, y_i) \in \mathcal{D}_{\text{train}} \mid y_i = c\}$.
**Output:** Episodic loss $\mathcal{L}(\theta)$ for sampled episode $e$.
{Select $N$ classes for episode $e$}
$C \leftarrow RandomSample(\{1, ..., M\}, N)$
{Compute cross-modal prototypes}
**for** $c$ in $C$ **do**
    $\mathcal{S}_e^c \leftarrow RandomSample(\mathcal{D}_{\text{train}}^c, K)$
    $\mathcal{Q}_e^c \leftarrow RandomSample(\mathcal{D}_{\text{train}}^c \setminus \mathcal{S}_e^c, K_Q)$
    $\mathbf{p}_c \leftarrow \frac{1}{|\mathcal{S}_e^c|} \sum_{(s_i, y_i) \in \mathcal{S}_e^c} f(s_i)$
    $\mathbf{e}_c \leftarrow LookUp(c, \mathcal{W})$
    $\mathbf{w}_c \leftarrow g(\mathbf{e}_c)$
    $\lambda_c \leftarrow \frac{1}{1 + \exp(-h(\mathbf{w}_c))}$
    $\mathbf{p}_c' \leftarrow \lambda_c \cdot \mathbf{p}_c + (1 - \lambda_c) \cdot \mathbf{w}_c$
**end for**
{Compute loss}
$\mathcal{L}(\theta) \leftarrow 0$
**for** $c$ in $C$ **do**
    **for** $(q_t, y_t)$ in $Q_e^c$ **do**
        $\mathcal{L}(\theta) \leftarrow \mathcal{L}(\theta) + \frac{1}{N \cdot K}[d(f(q_t), \mathbf{p}_c')) + \log\sum_k \exp(-d(f(q_t), \mathbf{p}_k'))]$
    **end for**
**end for**

## B  Descriptions of data sets

*mini*ImageNet.  This dataset is a subset of ImageNet ILSVRC12 dataset [40]. It contains 100 randomly sampled categories, each with 600 images of size $84 \times 84$. For fair comparison with other methods, we use the same split proposed by Ravi et al. [38], which contains 64 categories for training, 16 for validation and 20 for test.

*tiered*ImageNet.  This dataset is a larger subset of ImageNet than *mini*ImageNet. It contains 34 high-level category nodes (779,165 images in total) that are split in 20 for training, 6 for validation and 8 for test. This leads to 351 actual categories for training, 97 for validation and 160 for the test. There are more than 1,000 images for each class. The train/val/test split is done according to their higher-level label hierarchy. According to Ren et al. [39], splitting near the root of ImageNet hierarchy results in a more realistic (and challenging) scenario with training and test categories that are less similar.

CUB-200.  Caltech-UCSD-Birds 200-2011 (CUB-200) [55] is a fine-grained and medium scale dataset with respect to both number of images and number of classes, *i.e.* 11,788 images from 200 different types of birds annotated with 312 attributes [58]. We chose the split proposed by Xian *et al.* [58]. We used the 312-dimensional hand-crafted attribution as the semantic modality for fair comparison with other published modality alignment methods.

Word embeddings.  We use GloVe [37] to extract the semantic embeddings for the category labels. GloVe is an unsupervised approach based on word-word co-occurrence statistics from large text corpora. We use the Common Crawl version trained on 840B tokens. The embeddings are of dimension 300. When a category has multiple (synonym) annotations, we consider the first one. If the first one is not present in GloVe's vocabulary we use the second. If there is no annotation in GloVe's vocabulary for a category (4 cases in *tiered*ImageNet), we randomly sample each dimension of the embedding from a uniform distribution with the range (-1, 1). If an annotation contains more

than one word, the embedding is generated by averaging them. We also experimented with fastText embeddings [17] and observed similar performances.

## C  Baselines

For modality alignment baselines, we follow CADA-VAE [44]'s few-shot experimental setting. During training, we randomly sample $N$-shot images for the test classes, and add them in the training data to train the alignment model. During test, we compare the image query and the class embedding candidates in the aligned space to make decisions as in ZSL and GZSL.

For the meta-learning extensions of modality alignment methods, instead of including the $N$-shot images into training data, we follow the standard episode training (explained in Section 3) of metric-based meta-learning approach and train models only with samples from training classes. Moreover, during training, we add an additional loss illustrated in Equation 1 and 3, to ensure the metric space learned on the visual side matching the few-shot test scenario. At test, we employ the standard few-shot testing approach (described in Appendix D) and calculate the prototype representations of test classes as follows:

$$\mathbf{p}_c = \frac{\Sigma_i \mathbf{r}_i^c + \mathbf{w}_c}{N + 1}, \tag{7}$$

where $\mathbf{r}_i$ is the representation of the $i$-th support image. For both training and test, we need a visual representation space to calculate prototype representations. For DeViSE, they are calculated in its visual space before the transformer [9]. For both ReViSE and CADA-VAE, prototype representations are calculated in the latent space. For f-CLSWGAN, they are calculated in the discriminator's input space.

## D  Implementation Details of AM3 Experiments

We model the visual feature extractor $f$ with a ResNet-12 [12], which has shown to be very effective for few-shot classification [35]. This network produces embeddings of dimension 512. We use this backbone in all the modality-alignment baselines mentioned above and in AM3 implementations (with both backbones). We call *ProtoNets++* the prototypical network [47] implementation with this more powerful backbone.

The semantic transformation $g$ is a neural network with one hidden layer with 300 units which also outputs a 512-dimensional representation. The transformation $h$ of the mixture mechanism also contains one hidden layer with 300 units and outputs a single scalar for $\lambda_c$. On both $g$ and $h$ networks, we use ReLU non-linearity [10] and dropout [49] (we set the dropout coefficient to be 0.7 on *mini*ImageNet and 0.9 on *tiered*ImageNet).

The model is trained with stochastic gradient descent with momentum [51]. We use an initial learning rate of 0.1 and a fixed momentum coefficient of 0.9. On *mini*ImageNet, we train every model for 30,000 iterations and anneal the learning rate by a factor of ten at iterations 15,000, 17,500 and 19,000. On *tiered*ImageNet, models are trained for 80,000 iterations and the learning rate is reduced by a factor of ten at iteration 40,000, 50,000, 60,000.

The training procedure composes a few-shot training batch from several tasks, where a task is a fixed selection of 5 classes. We found empirically that the best number of tasks per batch are 5,2 and 1 for 1-shot, 5-shot and 10-shot, respectively. The number of query per batch is 24 for 1-shot, 32 for 5-shot and 64 for 10-shot. All our experiments are evaluated following the standard approach of few-shot classification: we randomly sample 1,000 tasks from the test set each having 100 random query samples, and average the performance of the model on them.

All hyperparameters were chosen based on accuracy on validation set. All our results are reported with an average over five independent run (with a fixed architecture and different random seeds) and with 95% confidence intervals.

## E  Results on CUB-200

We also conduct experiments on CUB-200 to better compare with modality-alignment baselines from ZSL. Table 3 shows the results. For 0-shot scenario, AM3 degrades to the simplest modality

alignment method that maps the text semantic space to the visual space. Therefore, without the adaptive mechanism, AM3 performs roughly the same with DeViSE, which indicates that the adaptive mechanism play the main role on the performance boost we observed in FSL. The results on other few-shot cases on CUB-200 are consistent with the other two few-shot learning data sets.

We also conduct generalized few-shot learning experiments as reported for CADA-VAE in [44] to compare AM3 with the published FSL results for CADA-VAE. Figure 4 shows that AM3-ProtoNets outperforms CADA-VAE in every case tested. We consider as a metric the harmonic mean (H-acc) between the accuracy of seen and unseen classes, as defined in [56, 44].

| Model | Test Accuracy | | |
|---|---|---|---|
| | 0-shot | 1-shot | 5-shot |
| DeViSE [9] | 52.0% | 54.7% | 60.4% |
| ReViSE [14] | 55.2% | 56.3% | 63.7% |
| VZSL [] | 57.4% | 60.8% | 70.0% |
| CBPL [29] | 61.9% | - | - |
| f-CLSWGAN [57] | 62.1% | 64.7% | 73.7% |
| CADA-VAE [44] | 61.7% | 64.9% | 71.9% |
| ProtoNets | - | 68.8% | 76.4% |
| AM3-ProtoNets | 51.3% | 73.6% | **79.9**% |
| TADAM [35] | - | 69.2% | 78.6% |
| AM3-TADAM | 50.7% | **74.1**% | 79.7% |

Table 3: Few-shot classification accuracy on *unseen-test* split of CUB-200.

Figure 4: H-acc of generalized few-shot learning on CUB-200.

# F    Ablation study on the input of the adaptive mechanism

We also perform an ablation study to see how the adaptive mechanism performs with respect to different features. Table 4 shows results, on both datasets, of our method with three different inputs for the adaptive mixing network $h$: (i) the raw GloVe embedding ($h(\mathbf{e})$), (ii) the visual representation ($h(\mathbf{p})$) and (iii) a concatenation of both the query and the language embedding ($h(\mathbf{q}, \mathbf{w})$).

We observe that conditioning on transformed GloVe features performs better than on the raw features. Also, conditioning on semantic features performs better than when conditioning on visual ones, suggesting that the former space has a more appropriate structure to the adaptive mechanism than the latter. Finally, we note that conditioning on the query and semantic embeddings helps with the ProtoNets++ backbone but not with TADAM.

| Method | ProtoNets++ | | TADAM | |
|---|---|---|---|---|
| | 1-shot | 5-shot | 1-shot | 5-shot |
| $h(\mathbf{e})$ | 61.23 | 74.77 | 57.47 | 72.27 |
| $h(\mathbf{p})$ | 64.48 | 74.80 | 64.93 | 77.60 |
| $h(\mathbf{w}, \mathbf{q})$ | 66.12 | 75.83 | 53.23 | 56.70 |
| $h(\mathbf{w})$ (AM3) | 65.21 | 75.20 | 65.30 | 78.10 |

Table 4: Performance of our method when the adaptive mixing network is conditioned on different features. Last row is the original model.