[Reviews · NeurIPS 2019]

Reviewer 1



Pros: — Paper is overall well-written — Approach is simple yet effective — Method achieves state of the art (SOTA) performance on standard benchmarks — Overall, this is a solid paper which warrants acceptance Cons: — Some expositional details should be added (see details) — Some additional ablations would be useful (see details) Detailed Comments ----------------------- > Exposition Overall, the paper provides a clear introduction that motivates use of cross-modal information and lays out the high-level overview of the contributions. The experiments are also well structured and presented clearly. That said, some improvements are possible and would be warranted: — The description of the problem at the beginning of Section 3.1.1 was confusing at first glance. Given the variety of few-shot learning scenarios, such as generalized few-shot learning, use of better notation and formalities in laying out the support and query sets more mathematically (as opposed to descriptively) would be useful. — A more detailed overview of TADAM, the second model being extended (in addition to ProtoNets), was missing. This should be added. — The ZSL case is mentioned in multiple places throughout the work, where at times seems unnecessary. > Novelty Method is conceptually simple, yet practically effective (as shown by the new SOTA performance achieved on few-shot image classification on the miniImageNet and tieredImageNet datasets). The paper's adaptive convex combination using learned transformation and mixing networks is interesting and relatively novel. > Significance The fact that the method is agnostic with respect to which metric-based model it is extending is a positive. As such, the proposed model may have potential of extending uni-modal metric based models using other modes of information, such as class definitions and knowledge graphs. Seemingly it can also potentially be applied more broadly in other few-shot learning tasks. > Experiments The experiments on few-shot image classification are reasonable in scope and coverage. The chosen baselines cover the majority of SOTA and near-SOTA methods previously proposed with good coverage across meta-learners, metric-based methods, space aligners, variational methods and other approaches. The CUB-200 results, although mainly included in the supplemental material, also support the same conclusion and the reported results show the superior performance of the proposed method on a variety of k-shot 5-way tasks. — It would have been useful to also report results on MNIST although presumably the bi-modal class embeddings for the digits may have not been as effective. — Additional ablations should be added. Specifically on the"N-way" behavior of the method. Are similar improvements observed with more classes? Although 5-way classification setting is most widely used, the number of few-shot variables and the behavior of the method with respect to such factor is important. > POST REBUTTAL Authors have largely addressed my comments. While simple, the approach appears effective and valuable. I continue to be in favor of paper's acceptance.

Reviewer 2



Post Rebuttal: Thank you to the authors for their comments. My concern was primarily about not having labels for classes in the test-episodes, since few-shot learning models can be applied to learn about arbitrary classes at evaluation without requiring any syntactic labels. With regard to the author's comments on simplicity of the model and being applicable to any metric-based method, I have increased my score. Before Rebuttal: This paper proposes a modification to the few-shot learning image classification scenario, which involves using additional cross-modal semantic information in the form of word embeddings for class labels. The idea is that the extra semantic information can be helpful in a lot of cases where discriminating between visually-similar classes can be difficult, especially when given few examples. The authors propose an extension of prototypical networks for this setting where a prototype is the convex combination between the usual prototype computed from support item features and a learned transformation of the word embedding of the class label. The coefficient given to each part of the sum is a learned function of the word embedding. The method is evaluated on two few-shot learning benchmarks: Mini-Imagenet and Tiered-Imagenet. Additional studies are conducted to study the benefit of the extra semantic features as a function of increasing number of examples for few-shot learning. Strengths 1. Paper is well-written - motivates and describes idea well. 2. Extensive comparison of relevant baselines. Weaknesses 1. Proposed method is relatively simple extension - involves using typical prototype for class in addition to transformation of class word-embeddings. 2. The benefit of incorporating semantic information is largely in the 5-class, 1-shot learning case (3.5% Mini-Imagenet and 2.75% Tiered-Imagenet accuracy gain compared to state-of-the-art LEO applied to regular few-shot learning scenario) and there seems to be very little gain beyond that number of shots. Comments The proposed setting and method requires that word embeddings are known for all train and test classes. Is it a more realistic scenario for few-shot learning that word-embeddings are only available for train classes, as this removes requirement that model that can only be used to learn about concepts that we already have word-embeddings for? Is semantic information as defined in paper applicable to few-shot settings beyond image-classification?

Reviewer 3



1. Although the gated fusion of visual and semantic representation is simple, but the idea is interesting and effective. 2. To my knowledge, I have not see similar idea in few-shot learning. But I am not sure if there might be other work having similar ideas or implementations? 3. In the gated formulation Equation 5, the computation of \lambda should come from the concatenation of both visual representation and word embedding rather than only word embedding. Please refer to the following two papers: 1, Gated fusion unit for multimodal information fusion, ICLR. 2, Learning semantic concept and order for image and sentence matching, CVPR.

[Author Response · NeurIPS 2019]

We thank reviewers for detailed and helpful reviews. We particularly acknowledge that reviewers find this work effective
(R1, R2, R3), well written (R1, R2), interesting (R3) and achieves good results (R1, R3). Next, we address the main
concerns from reviewers.

**Presentation and extra ablation studies (R1)**. We thank R1 for pointing some expositions issues and the proposed
improvements will be added to final version of the manuscript. We conduct 10-way-1-shot, 10-way-5-shot experiments
on *mini*ImageNet to verify that the performance improvement of AM3 is consistent when the number of classes
change. Table 1 shows the results. Due to time limits, we only compared AM3 with its backbone methods and the
modality alignment method that performed the best in 5-way scenarios. Results show that AM3 also outperforms its
backbone methods and the modality-alignment baseline in 10-way setting. After rebuttal, we will test all baselines on
*mini*ImageNet 10-way setting to make a thorough comparison and try more variance of N-way FSL settings.

**Realistic scenario (R2).** We argue that cross-modal few-shot learning sce-
nario (ie, having access to both train and test word embeddings) is realistic.
If we understand correctly, R2's main concern is that the word embeddings of
the test labels may not be accessible. We believe that it would hardly happen.
The reasons are as follows.

First, GloVe (chosen as pretrained word embeddings) is trained *unsupervised*
and contain embeddings for 1.9M words. It is likely that a semantic label will
lie in the GloVe vocabulary. Even if it is not the case, it would still be realistic
to simply crawl considerably amount of text from the web that contains the
token of the test label, to (unsupervised) train a word embedding model using
the same technology. Therefore, as long as we have access to the labels of

| Model | 1-shot | 5-shot |
|---|---|---|
| CADA-VAE-FSL | 37.5% | 56.3% |
| ProtoNets++ | 39.1% | 59.5% |
| AM3-ProtoNets++ | 45.7% | 61.4% |
| TADAM | 42.7% | 61.2% |
| AM3-TADAM | 47.3% | 62.1% |

Table 1: 10-way classification accuracy.

the test set, getting meaningful word embeddings for them without any supervision (human labeling efforts) is relatively
straight forward. Second, we can easily assume a FSL scenario in which we have access to the labels of the test set.
In vanilla FSL, a support set of "few-shot" samples is provided for each unseen category. In our scenario, we assume
the label of each support set is also given (eg, images of cat and the semantic label 'cat'). We found this a realistic
assumption. Third, leveraging word embeddings (trained on unlabeled text corpora) of class labels (for both train and
test classes) for vision tasks has long been exploited (eg, ZSL, image retrieval, image captioning, etc.)

**"Method is a simple extension of prototypical networks" (R2).** We disagree. On the contrary, AM3 is model-
agnostic to any metric-based FSL methods, as described in the paper. In experiments, we test AM3 on two different
metric-based FSL: ProtoNets and TADAM. We agree with R1 that "the fact that the method is agnostic with respect to
which metric-based model it is extending is a positive". Moreover, we think the "extension" is not simple. Although
employing extra knowledge source to help FSL may sound straightforward, the cross-modal method should be designed
to fit FSL scenario – a task that is not trivial. Existing complicated cross-modality models (modality-alignment methods
and proposed baselines) fail to work well in FSL, as our experiments show. The main contribution of our paper is the
model that is designed to conduct cross modality specifically in FSL scenario. We empirically demonstrated it to be
effective at integrating extra information from unsupervised text corpus to boost performance on the few-shot image
recognition task.

**Simplicity of the model (R2).** R2 points the simplicity of the model as a weakness. We share the opinion of R1 and R3,
mentioning this work is "simple yet effective" and "interesting and effective". Most of the modality alignment baselines
we compared are quite complicated (eg, CADA-VAE employs 2 VAEs). However, due to their assumption that the two
modalities have to be aligned (too rigid for FSL, as argued on the paper), their performances can't outperform AM3. A
model as simple as AM3 outperforms complicated baselines to a large margin. In this circumstance, we disagree that
the simplicity of AM3 is its weaknesses.

**Application beyond image classification (R2).** As pointed by R1 and R3, the proposed approach can potentially
be used in many different cross-modal FSL settings involving visual and semantic information. Few-shot semantic
segmentation, object detection, action recognition, etc, can be some of them.

**Gated formulation in Eq. 5 (R3).** We thank R3 for the suggestion on the input of the gated formulation in Eq. 5. We
agree that intuitively it would make more sense if $\lambda$ is conditioned by both variables. We will empirically verify it after
the rebuttal and update the paper accordingly. We will also discuss the differences wrt the papers mentioned by R3.

**"Any other work having similar ideas or implementations?" (R3).** To the best of our knowledge, AM3 is the first
model in FSL setting that proposes a gated fusion of representations of the two modalities. Other models that incorporate
cross-modal information in low-data regime (eg, ZSL and FSL) are based on modality-alignment methods. As argued
on the submission, modality-alignment methods force the two spaces to have the same semantic structure, which is too
rigid for FSL, given that we have supports from the original modality at test.

[Meta-Review · NeurIPS 2019]

The paper presents a simple yet effective method for cross-modal few-shot learning, with compelling results on standard benchmarks. All reviewers recommend acceptance, and the AC agrees with this decision.